

**Prediction of geomorphologic parameters of catchment without GIS to**
**estimate runoff using GIUH model**
Pouyan Keshtkaran[1*], Touraj Sabzevari[2]
[1] Department of civil engineering, Estahban Branch, Azad University, Estahban, Iran
* Corresponding Author: e-mail: water.estahban@yahoo.com
Tel: 00989173132092
Fax: 009871132360352
[2] Department of civil engineering, Estahban Branch, Azad University, Estahban, Iran
e-mail: tooraj419@yahoo.com
Tel: 00989177151596
**Abstract:**
Estimation of flood in ungauged catchments has great importance in the design of hydraulic
structures. The geomorphologic instantaneous unit hydrograph (GIUH) technique uses
geomorphologic parameters to estimate catchment runoff. In this research, regression equations
were developed based on geometrical characteristics of nine catchments such as area, length and
slope of the main river to estimate geomorphologic data of other catchments with no need for
GIS and digital elevation model. These equations were used for verification of stream-order-law
ratios as well as geomorphologic parameters corresponding to the Gagas, Heng-Chi and Kasilain
catchments. In this study, the effect of stream-order-law ratios on the rate of runoff in Kasilian
catchment was examined, and the sensitivity of each ratio was analyzed. The GIUH model was
assessed in two cases of GIS-supported and GIS-unsupported. The mean errors of the regression
equations in estimation of ratios $R_B$, $R_L$, $R_A$, $R_S$ and $R_{SO}$ in three study catchments were 4.7%,
23.5%, 7.1%, 41.3%, and 22.9%, respectively. The direct runoff hydrograph for the Heng-Chi
and the Kasilian catchments were computed by GIUH model and compared with observed



runoff. According to the results, the errors of peak discharge for four rainfall-runoff events in
GIS-unsupported case were, on average, 10% more than the error in the case of GIS-supported
GIUH. The results of GIUH for the two cases are very close to each other. The mean coefficient
of efficiency of the model was computed as 0.87.
**Key words:** GIUH, GIS, Stream-order-law ratios, Geomorphologic parameters
**1. Introduction**
Estimation of design flood in catchments is a vital issue in design of flood control structures.
Most catchments are ungauged and the statistical methods are not efficient, hence the rainfall-
runoff models are employed to estimate runoff. GIUH is a rainfall-runoff model for estimating
runoff in ungauged catchments using their geomorphologic parameters (GP).
Studies on streams orderings of catchments were first introduced by Horton (1932, 1945). Later,
modifications were made on Horton's method by Strahler (1952, 1957, 1964) leading to a new
method of ordering.
The idea of GIUH was introduced by Rodriguez-Iturbe and Valdes (1979). They suggested an
instantaneous unit hydrograph (IUH) model in which time to peak and peak flow of the
catchment were functions of geomorphologic features. The geomorphologic parameters of the
catchments are calculated by GIS software such as ArcGis and hydrologic extensions such as
ArcHydro. For this purpose, DEM of the catchment is necessary. First, stream networks are
delineated and, GP such as the number of streams, lengths, slopes, and drainage areas in each
order of streams is carried out based on stream orderings (Horton-Strahler method).  GIUH
model was extended and used by other scientists in different catchments (e.g. Gupta et al. 1980;
Rodriguez-Iturbeet al. 1982; Lee and Yen 1997 and Kumar and Kumar 2008).
An alternative approach was provided by Lee and Yen (1997). The travel times for different
orders of overland areas and channels were derived using the kinematic-wave theory and then
substituted into the GIUH model to develop a kinematic wave-based GIUH model for watershed
runoff simulation.
Lee and Chang (2005) offered a GIUH model to estimate surface and subsurface flow of
catchments. In their research, special importance was given to separation of surface flow from
subsurface flow in catchments. Sabzevari et al. (2013) modified the model presented by Lee and



Chang (2005) for estimation of surface and subsurface flow of Kasilian catchment. They have
also given a saturation model for separation of saturated and unsaturated zones of overland
regions.
Sabzevari and Norouzpoor (2014) suggested a GIUH model which is capable of taking plan
shape and profile curvature in complex hillslopes in computation of surface and subsurface travel
time. Also, the effect of geometry of complex hillslopes on the runoff in sub-catchment No. 125
of Walnut Gulch was investigated.
Kumar et al. (2004, 2007) rendered the runoff estimation of ungauged catchments by applying
the GIUH-based Nash and Clark models. They used stream ratios to estimate Nash and Clark's
parameters. Kumar and Kumar (2008) focused on estimation of runoff in Ramganga catchment,
India, applying GIUH based on kinematic wave theory. The model was used in the cases where
the inputs were geomorphologic parameters and stream-order-law ratios. Travel time of the
streams and overland regions in the two above cases were given as analytic equations based on
Horton-Strahler stream-ordering system.
Choi et al. (2011) used a concept of geomorphologic dispersion to estimate Nash model
parameters from spatial heterogeneity of flow path within a catchment.
Based on GP of catchment, stream-order-law ratios such as bifurcation ratio ($R_B$), stream-length
ratio ($R_L$), stream-area ratio ($R_A$), and stream-slope ($R_S$) ratio could be computed. According to
the GIUH offered by Yen and Lee (1997), the travel times of overland region and stream could
be worked out regarding stream-order-law ratios prior to IUH estimation.
Due to the lack of topographic map and DEM for most of the catchments, application of GIS-
based GIUH models is practically useless. One goal of this research is to provide a technique by
which one could compute geomorphologic parameters without the need for GIS. Calculating the
GP by means of GIS is costly and takes a long time. For example, extensions such as ArcHydro,
though capable of calculating the number, length, and slope of streams at any order, provide no
information about overland surface slopes or drainage area at any order which ought to be
calculated manually by GIS specialists which is time consuming. For this purpose, GP of twelve
catchments of various sizes with diverse stream networks were collected. The values of stream-
order-law ratios and the actual GP of the catchments obtained from GIS were derived.



To study the relation between data, linear and nonlinear regressions were used using the SPSS
software. In general, length and slope of the main stream and area of the catchment are among
the geometric parameters that are easily computable for every catchment. It is also important to
present empirical equations which could predict all stream-order-law ratios based on the
geometrical catchment information.
The important aims of this research are:
(1) to present equations which can predict, without the use of GIS and DEM of the catchment,
the stream-order-law ratios on the basis of length, slope of the main stream and area of
catchment (geometrical features).
(2) to analyze sensitivity of stream ratios and its effect on direct runoff hydrograph (DRH).
(3) to estimate runoff of ungauged catchments by means of GIUH without the use of GIS.

**2. GIUH model**

Surface runoff of the overland regions moves, through stream networks, to the outlet of
catchment. If a catchment is ordered via Strahler ordering scheme, the water travel paths from
the overland regions to the outlet are specified. Each flow path is comprised of different states,
the first of which is the overland region and the others are the streams. The probability of water
motion in a certain path $w: x_{o_i} \rightarrow x_i \rightarrow x_j \rightarrow ... \rightarrow x_\Omega$ is expressed as:

$$P(w) = P_{OA_i} P_{x_{oi} x_i} P_{x_i x_j} ... P_{x_k x_\Omega} \qquad (1)$$

where $P_{OA_i}$ is the initial state probability of rain drop moving from $i$th order overland region to
the $i$th order stream, which can be approximated as the ratio of $i$th order overland area to the total
catchment area; $P_{x_{oi} x_i}$ which is the probability of raindrop moving from $i$th order overland region
$(x_{o_i})$ to $i$th order stream equals one; and $P_{x_i x_j}$ is the transitional probability of rain drop moving
from $i$th order stream $(x_i)$ to $j$th order channel $(x_j)$.
The number of streams at each order and how they are connected to each other specify the
probabilities in Eq. (1).



The value of IUH of a watershed comprising different runoff paths is given by Eq. (2)
(Rodriguez-Iturbe and Valdes 1979).
$$u(t) = \sum_{w \in W} [f_{x_{o_i}}(t) * f_{x_i}(t) * f_{x_j}(t) * .... * f_{x_\Omega}(t)]_w \times P(w) \qquad (2)$$
where $f_{x_k}(t)$ denotes the travel time probability density function (PDF) in state $x_k$ with a mean
travel time value $(T_{x_k})$ and the function $f$ is indeed the IUH of any state $x_k$ calculated by the
formula $f(t) = (1/T_{x_k}) \exp(-t/T_{x_k})$. The PDF is a function of the travel time of each state in the
overland regions and streams. Asterisk (*) denotes a convolution integral. $w \in W$, $W$ being
$W = \langle x_{o_i}, x_i, x_j, ..., x_\Omega \rangle$, $i = 1, 2, 3, ..., \Omega$ and $t$ is the time.
To solve Eq. (2), one could resort to the Laplace transformations. In the process of GIUH
derivation, computation of travel time is the most intricate part of the work because its value
depends on GP of the catchment.
The ordinates of DRH for the catchment were estimated by convoluting the effective rainfall
hyetograph with the derived IUH.
The equation for estimation of DRH is:
$$Q(t) = \int_0^t u(t-\tau) I_e(\tau) d\tau \qquad (3)$$
where $I_e$ is the excess rainfall and $u(t)$ is the catchment IUH.
**2.1. Travel time of overland planes and streams**
According to the kinematic wave theory, the travel time of an overland plane depends on the
length, slope, Manning coefficient, and excess rainfall intensity. Eq. (4) which is due to Yen and
Lee (1997) gives the travel time of the $i$th overland plane.
$$T_{X_{oi}} = \left( \frac{n_0 A P_{OA_i} \sum_{i=1}^{\Omega} R_L^{i-\Omega}}{2 a^{1/2} S_{c_\Omega}^{b/2} L q_L^{m-1} R_B^{\Omega-i} R_L^{i-\Omega} R_S^{b(i-\Omega)/2}} \right)^{1/m} \qquad (4)$$


where $R_B$, $R_L$, $R_A$, and $R_S$ are bifurcation ratio, stream-length ratio, stream-area ratio, and stream-
slope ratio, respectively; $A$ is the area of the catchment; $a$ and $b$ are 5.463 and 1.083,
respectively; $q_L$ is the excess rainfall intensity; $n_0$ is the Manning's roughness coefficient for
overland flow; $S_{c\Omega}$ is slope of the highest order stream; the constant $m$ can be recognized as 5/3
from Manning's equation and $L$ is sum of mean length of the streams of different orders.
The travel time of the $i$th-order channel in each path is obtained, based on its GP, through Eq. (5)
(Yen and Lee 1997):

$$T_{X_i} = \frac{B_\Omega L R_L^{i-\Omega} R_B^{\Omega-i} \sum_{i=1}^{i} R_L^{i-\Omega}}{q_L A P_{OA_i} (\sum_{i=1}^{\Omega} R_L^{i-\Omega})^2} \left[ \left( h_{co_i}^m + \frac{q_L A P_{OA_i} n_c \sum_{i=1}^{\Omega} R_L^{i-\Omega}}{B_\Omega S_{c\Omega}^{1/2} R_S^{(i-\Omega)/2} R_B^{\Omega-i} \sum_{i=1}^{i} R_L^{i-\Omega}} \right)^{1/m} - h_{co_i} \right] \qquad (5)$$

where $h_{co_i}$ is the inflow depth of the $i$th-order channel due to water transported from upstream
reaches, is given as:

$$h_{co_i} = \left( \frac{q_L n_c A (R_B^{\Omega-i} R_A^{i-\Omega} - P_{OA_i}) \sum_{i=1}^{\Omega} R_L^{i-\Omega}}{S_{c\Omega}^{1/2} B_\Omega R_S^{(i-\Omega)/2} R_B^{\Omega-i} \sum_{i=1}^{i} R_L^{i-\Omega}} \right)^{1/m}$$

142    (6)

Where $n_c$ is the Manning coefficient of stream, $B_\Omega$ is the width of the stream. The value of $h_{coi}$ is
equal to zero for $i$=1.

**3. Geomorphologic parameters (GP)**
As observed in the Eqs. (5) and (6), the stream-order-law ratios particularly, $R_S$, $R_A$, $R_L$, $R_B$ are of
high importance. These affect the travel time, IUH, and DRH; also, they are computed according
to the GP. For this purpose, the stream network is delineated by means of GIS. In the GIS, the
streams are ordered via Horton-Strahler method, and the number, length, and slope of the
streams are calculated at each order.
The value of $R_B$ is given by the following equation regarding the number of stream segments at
each order:



$$R_B = N_{i-1} / N_i \quad ,i=2,3,...,\Omega \tag{7}$$
$N_i$ denotes the number of $i$th-order channels. The length ratio ($R_L$) is:
$$R_L = \overline{L_{c_i}} / \overline{L_{c_{i-1}}} \qquad ,i=2,3,...,\Omega \tag{8}$$

$\overline{L_{c_i}}$ is the mean length of $i$th-order channels. Eq. (9) yields the value of $R_A$:    157

$$R_A = \overline{A_i} / \overline{A_{i-1}} \tag{9}$$
where $\overline{A_i}$ is the mean area of catchment of order $i$. It should be noted that the mean area of a
given stream segment is, in fact, a cumulative value, for example, the area of a third-order
catchment is a sum of the areas of the first, second and third-order streams. Computation of $R_A$ is
not so easy a task for the GIS users.
The value of $R_S$ depends on the streams slope and is obtained by Eq. (10):
$$R_S = \overline{S_{c_i}} / \overline{S_{c_{i-1}}} \quad ,i=2,3,...,\Omega \tag{10}$$
where $\overline{S_{c_i}}$ is the mean slope of the $i$th-order streams.
As a result of experiments in the natural catchments, the following ranges are observed:
$3 \leq R_B \leq 5$ and $1.5 \leq R_L \leq 3.5$. Slope of the streams and overland planes for different catchments
at each order are different. The mean values of these slopes at each order take a considerable
time to compute by GIS, especially in large catchments.
In this research, a new slope ratio named the overland slope ratio ($R_{SO}$) is introduced that is given
in terms of the mean slope of the overland plane by:
$$R_{SO} = \overline{S_{o_{i-1}}} / \overline{S_{o_i}} \tag{11}$$
where $\overline{S_{o_i}}$ is the mean slope of the $i$th-order overland plane.
In this research we intend to find the relationship between $R_{SO}$ and the other stream-order-law
ratios.



Herein, a way for computing GP via regression equations is sought. These equations attained by
regression methods work through statistical analysis of the information of catchments possessing
geomorphologic attributes. The way these equations perform computations will be explained in
the next sections.
**4. Case Study**
To study the relationship between geomorphologic parameters, knowledge of the GIS based GP
(i.e the GP derived from GIS) of some natural catchments is required. This research uses
information received from twelve catchments in different countries. Table (1) shows the GIS
based GP along with stream order ratios of the case study catchments. The catchments Long chi
(Shuyou et al. 2010); Long men (Shuyou et al. 2010); Chaukhutia (Kumar 2014); Al-Malaqi
(Shadeed et al. 2007); Debarwa (Alemngus and Mathur 2014); Gherghera (Alemngus and
Mathur 2014); San-Hsia (Chang and Lee 2008); Al-Badan (Shadeed et al. 2007); Al-Faria
(Shadeed et al. 2007) were used for training and estimation of regression equations, and the
Gagas (Kumar and Kumar 2008), Heng-Chi (Lee and Chang 2005) and Kasilian (Sabzevari et al
2013) catchments were used for verification of the suggested equations.
The columns Table (1)  (from left to right) illustrate, respectively, the catchment name, stream
order ($i$), number of streams, mean stream length, mean stream area, mean stream slope, mean
overland slope, $R_B$, $R_L$, $R_A$, $R_S$, and $R_{SO}$.
The Heng-Chi catchment is located in northern Taiwan and has an area of 53 km$^2$ (Lee 1998).
The Gagas catchment lies in the middle and outer range of the Himalayas in Uttarakhand State of
India and has an area of 506 km$^2$ (Kumar and Kumar 2008). The Kasilian Catchment is located
between 53° 18$^{'}$ E and 53° 30$^{'}$ E longitudes and 35° 58$^{'}$ N to 36° 7$^{'}$ N latitudes in the north of Iran
and has an area of 67.8 km$^2$. Figure (1) shows the Gagas and Kasilian catchments.









Table 1. GP of twelve case study catchments

| Catchment Name | Geomorphologic parameters | | | | | | | | | |
|---|---|---|---|---|---|---|---|---|---|---|
| | Order | $N_i$ | $\overline{L_i}$ | $\overline{A_i}$ | $\overline{S_c}$ | $\overline{S_o}$ | $R_B$ | $R_L$ | $R_A$ | $R_S$ | $R_{SO}$ |
| 1. Gagas | 1 | 121 | 1.74 | 3.02 | 0.172 | 0.810 | 4.8 | 2.4 | 5.4 | 0.4 | 2.6 |
| | 2 | 23 | 3.04 | 18.58 | 0.141 | 0.655 | | | | | |
| | 3 | 6 | 7.63 | 79.22 | 0.041 | 0.172 | | | | | |
| | 4 | 1 | 23.4 | 506 | 0.017 | 0.065 | | | | | |
| 2. Heng-Chi | 1 | 30 | 0.66 | 1.043 | 0.087 | 0.450 | 3.3 | 2.6 | 4 | 0.6 | 1.1 |
| | 2 | 6 | 2.74 | 6.919 | 0.050 | 0.419 | | | | | |
| | 3 | 2 | 1.6 | 19.9 | 0.012 | 0.349 | | | | | |
| | 4 | 1 | 4.97 | 53.23 | 0.012 | 0.347 | | | | | |
| 3. Kasilian | 1 | 42 | 1.6 | 0.915 | 0.241 | 0.345 | 3.5 | 1.5 | 4.3 | 0.4 | 1.1 |
| | 2 | 11 | 1.79 | 4.813 | 0.070 | 0.297 | | | | | |
| | 3 | 3 | 2.45 | 20.75 | 0.047 | 0.263 | | | | | |
| | 4 | 1 | 4.65 | 67.8 | 0.008 | 0.261 | | | | | |
| 4. San-Hsia | 1 | 69 | 0.92 | 1.15 | 0.161 | 0.314 | 4.2 | 2.9 | 5 | 0.4 | 1.1 |
| | 2 | 16 | 2.08 | 4.99 | 0.092 | 0.203 | | | | | |
| | 3 | 3 | 3.88 | 18.15 | 0.037 | 0.364 | | | | | |
| | 4 | 1 | 17.8 | 125.9 | 0.013 | 0.293 | | | | | |
| 5. Al-Badan | 1 | 41 | 1.38 | 1.37 | 0.170 | 0.140 | 4 | 1.5 | 4.5 | 1 | 1.7 |
| | 2 | 6 | 3.2 | 10.12 | 0.092 | 0.062 | | | | | |
| | 3 | 2 | 5.03 | 40.73 | 0.140 | 0.051 | | | | | |
| | 4 | 1 | 3.17 | 85 | 0.135 | 0.029 | | | | | |
| 6. Al-Faria | 1 | 49 | 1.03 | 0.937 | 0.154 | 0.117 | 4 | 1.5 | 4.3 | 1.1 | 1.6 |
| | 2 | 8 | 2.12 | 6 | 0.085 | 0.058 | | | | | |
| | 3 | 3 | 3.5 | 19.4 | 0.161 | 0.033 | | | | | |
| | 4 | 1 | 2.62 | 64 | 0.125 | 0.031 | | | | | |
| 7. Al-Malaqi | 1 | 62 | 1.92 | 1.81 | 0.146 | 0.140 | 9 | 1.3 | 17 | 0.8 | 4.3 |
| | 2 | 16 | 2.61 | 5.83 | 0.122 | 0.063 | | | | | |
| | 3 | 1 | 3.21 | 185 | 0.081 | 0.010 | | | | | |
| 8. Debarwa | 1 | 23 | 2.26 | 5.6 | 0.032 | 0.135 | 4.9 | 3 | 6 | 0.6 | 1.2 |
| | 2 | 6 | 4.2 | 27.8 | 0.018 | 0.091 | | | | | |
| | 3 | 1 | 17.7 | 195 | 0.010 | 0.098 | | | | | |
| 9. Gherghera | 1 | 58 | 2.45 | 5.9 | 0.027 | 0.136 | 2.9 | 1.4 | 3.3 | 0.9 | 1.4 |
| | 2 | 14 | 4.19 | 30.6 | 0.018 | 0.087 | | | | | |
| | 3 | 5 | 10.2 | 101.0 | 0.010 | 0.064 | | | | | |
| | 4 | 2 | 4.47 | 259.9 | 0.016 | 0.025 | | | | | |





| | 5 | 1 | 4.19 | 525.7 | 0.011 | 0.117 | | | | | |
|---|---|---|---|---|---|---|---|---|---|---|---|
| 10. Long chi | 1 | 46 | 1.13 | 2.5 | 0.210 | 0.444 | 3.7 | 2.4 | 4 | 0.6 | 1.1 |
| | 2 | 10 | 3.45 | 11.8 | 0.124 | 0.487 | | | | | |
| | 3 | 4 | 3.19 | 32 | 0.073 | 0.514 | | | | | |
| | 4 | 1 | 9.94 | 141.8 | 0.054 | 0.364 | | | | | |
| 11. Long men | 1 | 58 | 1.31 | 2.74 | 0.560 | 0.256 | 4 | 2.2 | 4.7 | 0.9 | 1.8 |
| | 2 | 13 | 2.48 | 12.3 | 0.560 | 0.123 | | | | | |
| | 3 | 3 | 9.33 | 77.11 | 0.560 | 0.056 | | | | | |
| | 4 | 1 | 8.18 | 246.8 | 0.385 | 0.056 | | | | | |
| 12. Chaukhutia | 1 | 134 | 1.41 | 2.27 | 0.191 | 0.910 | 5.3 | 2.5 | 5.7 | 0.5 | 2.4 |
| | 2 | 31 | 2.65 | 12.28 | 0.123 | 0.567 | | | | | |
| | 3 | 7 | 7.21 | 60.18 | 0.041 | 0.174 | | | | | |
| | 4 | 1 | 20.7 | 452.3 | 0.019 | 0.074 | | | | | |



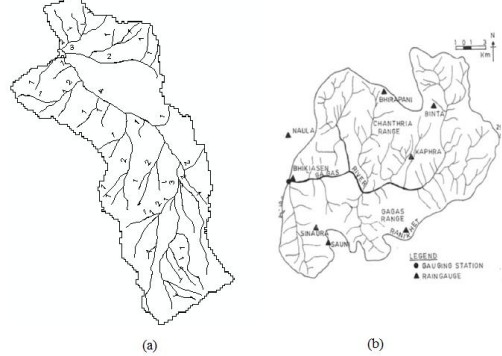

(a)                     (b)

Fig. 1 a) Kasilian catchment stream network   b) Gagas catchment          208


**5. Relationships of geomorphologic parameters**
**5.1. Estimation of bifurcation ratio ($R_B$)**
To estimate the bifurcation ratio of a catchment, the information concerning 80 watersheds with
areas between 1 km$^2$ and 600 km$^2$ were used which had known values of $R_B$ and area, with the
presumption that $R_B$ is a function of two variables, catchment area ($A$) and the main stream
length ($L$). With the help of SPSS18 software and using the information of 37 catchments an
optimum relation was obtained as:



$\quad R_B = 0.0027A + 3.47$ (12)
Admittedly, the value of $R_B$ was not dependent on L. The correlation coefficient of the fitted
equation is 0.8 and the real mean bifurcation ratio of the catchments is 4. Eq. (12) indicates that
in small catchments with area less than $200km^2$, the value of $R_B$ runs between 3.47 and 4. It is
suggested that Eq. (12) be applied to catchments of areas beneath $600km^2$. It should be noted
that, regarding Eq. (7) and $R_B$, the values of $N_i$ are calculated for $i \leq \Omega$. $\Omega$ is the maximum order
of the catchment. $N_{i=\Omega} = 1$ is considered and $N_{i-1} = R_B N_i$, $i \leq \Omega$.
**5.2. Computation of stream-length Ratio ($R_L$)**
To calculate the length ratio $R_L$, it was taken as a function of the main stream length and the
whole catchment area. The fitted regression equation for the nine selected catchments according
to Table (1) is, as follows:
$\quad R_L = 2.59 L^{0.41} A^{-0.2}$ (13)
The correlation coefficient is equal to 0.91. Based on Eq. (8) and $R_L$, the values of $\overline{L_{c_i}}$ are
calculated for $i \leq \Omega$. $\overline{L_{c_\Omega}} = L$ is considered and $\overline{L_{c_{i-1}}} = \overline{L_{c_i}} / R_L$, $i \leq \Omega$.
**5.3. Computation of area ratio ($R_A$)**
The area ratio was assumed to be a function of the bifurcation ratio and the length ratio with
fitted equation:
$\quad R_A = 0.597 R_B^{1.553} R_L^{-0.177}$ (14)
The correlation coefficient is 0.99. $\overline{A_\Omega} = A$ is considered and $\overline{A_{i-1}} = \overline{A_i} / R_A$, $i \leq \Omega$.
**5.4. Computation of stream slope ratio ($R_S$)**
Stream slope ratio was assumed to be a function of $R_B$, $R_L$, and $R_A$. Equation (15), having
correlation coefficient 0.79, represents the fitted regression relation for the data.
$\quad R_S = 1.198 R_B^{1.26} R_L^{-0.97} R_A^{-1.04}$ (15)





### 5.5. Computation of overland slope ratio ($R_{SO}$)

A nonlinear regression equation consisting of the parameters $R_B$, $R_L$, $R_A$, and $R_S$ was used to
calculate the slope ratio of the overland plane with the fitted relation:
$$R_{SO} = 0.366 R_B{}^2 R_L{}^{-0.58} R_A{}^{-0.66} \qquad (16)$$
The correlation coefficient of Eq. (16) is 0.93, and there is no strong correlation between $R_{SO}$ and
$R_S$. By the Eqs. (16) and (11) the slope of overland planes of the catchment could be obtained. It
is to be noted that the Eqs. (12) to (16) which are gained via the information about nine
catchments may be calibrated by adding more data. Given that the length of the main river and
the area in all catchments are known, the $R_B$, $R_L$, $R_A$, $R_S$, and $R_{SO}$ ratios can be calculated by Eqs.
(12) to (16).
The area of the catchment, the length and slope of the main river could be determined from the
simple topographic maps of the catchment. If a catchment has a maximum stream order $\Omega$, it is
inferred that the stream should be located at the end of the catchment with the mean slope $(\overline{S_{c_\Omega}})$
and the mean slope of the lateral overland planes $(\overline{S_{o_\Omega}})$. For instance, Fig. 2 shows a small
catchment with three subcatchment (I, II, III). The maximum stream order is two $(\Omega = 2)$. The
subcatchment III is created with two lateral overland planes and stream III is positioned at the
end of the main catchment. Fig. 2 shows the mean slope of the stream III $(\overline{S_{c_2}})$ and mean slope of
the two lateral overland planes $(\overline{S_{o_2}})$.

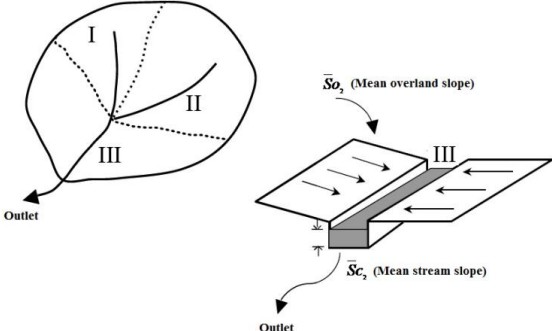






Fig. 2. Catchment with maximum stream order 2      260

If the values of the $\overline{S_{c_\Omega}}$ , $\overline{S_{o_\Omega}}$, $R_S$ and $R_{SO}$ are known , with regard to Eqs. (10) and (11), the
value of the $\overline{S_{c_i}}$ and $\overline{S_{o_i}}$ are computable for lower orders $i < \Omega$ ($\overline{S_{c_{i-1}}} = \overline{S_{c_i}} / R_S$ , $\overline{S_{O_{i-1}}} = \overline{S_{O_i}} R_{SO}$) .
**6. Effect of ratios $R_B$, $R_L$, $R_A$, $R_S$ and $R_{SO}$ on DRH**
In the previous section of this study, empirical equations were presented to obtain
geomorphologic ratios. Now, we apply the GIUH model to look into sensitivity analysis of these
ratios and their effects on DRH and on peak flood. To this end, the information of the Kasilian
catchment was utilized.
Fig. (3a) illustrates the effect of bifurcation ratio upon DRH of the Kasilian catchment on 4[th]
May, 1993.

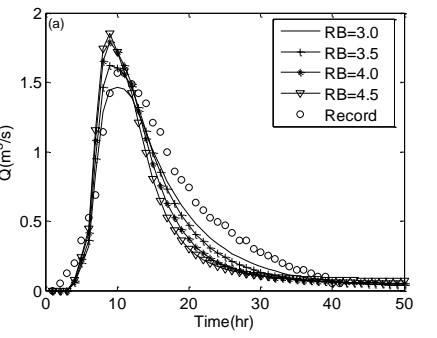
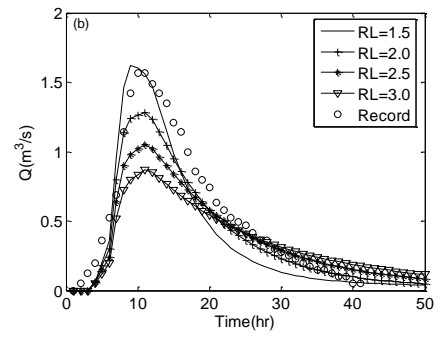

Fig. 3. Effect of $R_B$ and $R_L$ on DRH of the Kasilian catchment      271

The values of bifurcation coefficient 3, 3.5, 4, and 4.5 with 0.5 units increment were considered
for the Kasilian catchment, and the number of streams and the values of input parameters into
GIUH model were computed and inserted to the model. The effect of $R_B$ on shape of hydrograph
and peak of the runoff is seen in Fig. (3a).The results of the model are compared with those of
recorded runoff hydrographs.
To determine the effect of different values of $R_B$ on the peak of runoff, the following equation of
relative sensitivity was used:



$$S_r = \frac{O_2 - O_1}{P_2 - P_1}(\overline{P}/\overline{O})$$  (17)
where $O$ and $P$ represent particular model outputs and parameters, respectively. So, $S_r$ gives the
percentage change in $O$ for a 1% change in $P$. $\overline{P}$ is given by $(P_1+P_2)/2$ and $\overline{O}$ is given by
$(O_1+O_2)/2$. Results confirmed that the least computational error in peak discharge relative to the
observed peak discharge was shown by $R_B$=3.5 with 3.5%. The actual $R_B$ for the Kasilian
catchment is also 3.5. The mean relative sensitivity of $R_B$ derived from Eq. (17) is 0.56.
Fig. (3b) shows the effect of $R_L$ on DRH of the Kasilian catchment. The values of this ratio were
taken as 1, 1.5, 2, and 2.5 with a 0.5 increment. According to the results, $R_L$=1.5 has given the
least error in peak discharge with 3.6% value. The actual $R_L$ of the catchment is 1.46, and the
mean relative sensitivity of $R_L$ amounts to 0.92. The larger the value of $R_L$, the higher peak error.
The runoff is affected more by length ratio relative to bifurcation ratio, a fact seen also in Fig.
(3). The next section of the paper was dedicated to the effects of area ratio on the peak of runoff.
The values of area ratio were regarded to be between 3 and 6 with 1 unit increment values.
Figure (4) depicts the effect of area ratio on DRH.

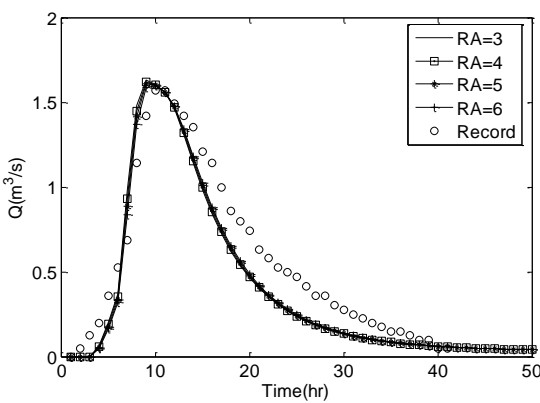


Fig.4. Effect of area ratio on DRH of the Kasilian catchment          294

As indicated by the results, the area ratio has had a slight effect on the runoff peak, so that
alterations of this ratio do not noticeably influence the shape of hydrograph and flood peak.
Fig. (5a) shows how $R_S$ affects DRH for the values 0.1, 0.4, 0.7, and 1 with a 0.3 increment.





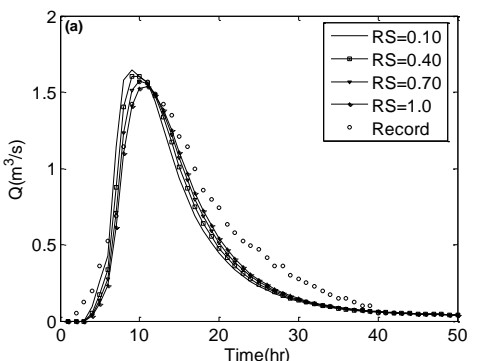 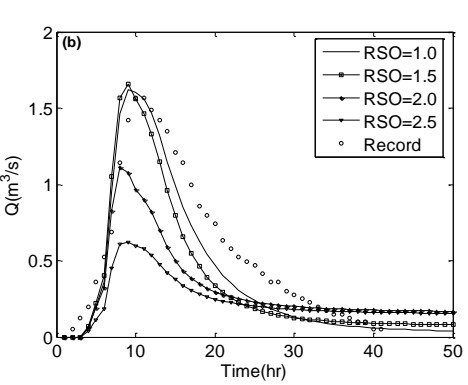


Fig. 5. Effect of $R_S$ and $R_{SO}$ on DRH of the Kasilian catchment    299

The least error is 0.47 which corresponds to the ratio (0.7) while the actual slope ratio of the Kasilian catchment is 0.38. Also, the mean relative sensitivity ratio is 0.042. The results indicate that this parameter has little effect on runoff peak, too.

Figure (5b) shows the influence of $R_{SO}$ on DRH for values of 1, 1.5, 2, and 2.5 with increment as 0.5. The least error relates to the ratio 1 which is 3.54%, whilst that of Kasilian catchment would be 1.1, and the mean relative sensitivity ratio 1.33. According to the results, the parameter $R_{SO}$ has remarkable effect on runoff peak.

According to the overall results, the relative sensitivity ratio of $R_B$, $R_L$, $R_A$, $R_S$, and $R_{SO}$ is 0.56, 0.92, 0.01, 0.042, and 1.33 respectively. The most effect concerns, correspondingly to the overland slope ratio, length ratio, bifurcation ratio, slope ratio, and area ratio.

To calculate the value of $P_{x_i x_j}$ in Eq. (1) the following equation is used:

$$P_{x_i x_j} = N_{i,j} / N_i \qquad (18)$$

where $N_{i,j}$ is number of $i$th order stream contributing the flow to $j$th order stream; $N_i$ is the number of $i$th order channel. The value of $N_i$ is computable by the bifurcation ratio, but to obtain the parameter $N_{i,j}$ the following equation is suggested:

$$N_{i,j} = 2N_i \exp(-0.64 j) \qquad (19)$$





which is obtained through nonlinear regression of the stream network data based on
geomorphologic parameters of the Kasilian and the Gagas catchments. In the catchments
possessing DEM one needs to delineate stream network and order them by GIS software,
however, calculation of $N_{i,j}$ should be done manually and rendered by GIS operator which is a
time-consuming and difficult task.
**7. Verification**
In the previous sections, equations were proffered for computation of stream-order-law ratios
based on GP in nine different catchments in the world. For verification of the results of the
regression equations the GP of three catchments Gagas, Heng-Chi, and Kasilian were applied.
Table (2) lists the GP as well as stream-order-law ratios of the three selected catchments using
Eqs. (12) to (16). The table (2) also provides the observed values of stream ratios and their
computational errors.

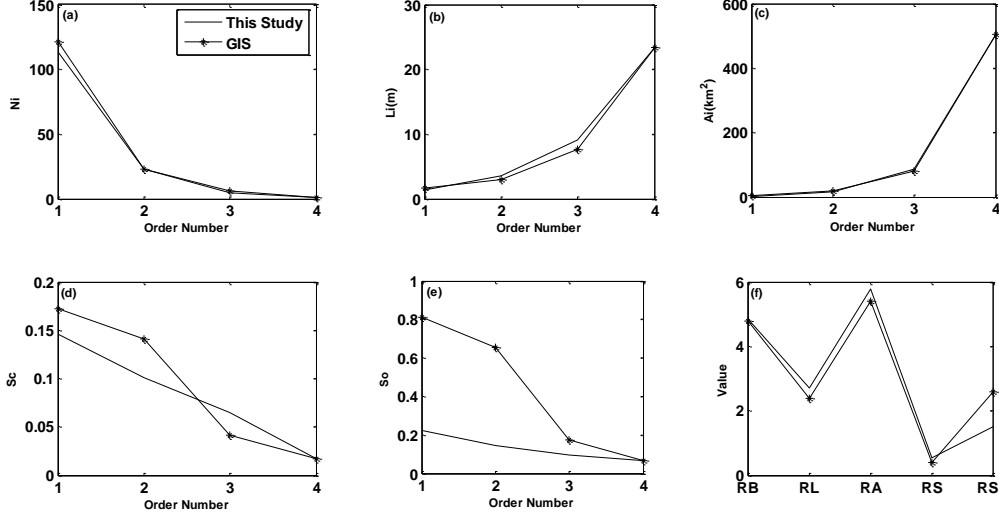


Fig. 6. Verification of GP in Gagas catchment





Table 2 Calculated GP of the Gagas, Heng-Chi, and Kasilian catchments          335

| Catchment Name | Geomorphologic parameters | | | | | | | | | | |
| --- | --- | --- | --- | --- | --- | --- | --- | --- | --- | --- | --- |
| | Order | $N_i$ | $\overline{L_i}$ | $\overline{A_i}$ | $\overline{S_c}$ | $\overline{S_o}$ | $R_B$ | $R_L$ | $R_A$ | $R_S$ | $R_{SO}$ |
| 1. Gagas | 1 | 113 | 1.38 | 2.6 | 0.146 | 0.222 | 4.84 | 2.72 | 5.78 | 0.53 | 1.5 |
| | 2 | 23 | 3.54 | 15.1 | 0.101 | 0.147 | | | | | |
| | 3 | 5 | 9.10 | 87.5 | 0.065 | 0.098 | | | | | |
| | 4 | 1 | 23.40 | 506.0 | 0.017 | 0.065 | | | | | |
| GIS Results | | | | | | | 4.80 | 2.40 | 5.40 | 0.40 | 2.60 |
| %Error | | | | | | | 0.40 | 13.7 | 7.6 | 21.0 | 41.4 |
| 2. Heng-Chi | 1 | 47 | 0.32 | 1.0 | 0.104 | 0.654 | 3.61 | 2.26 | 3.80 | 0.68 | 1.2 |
| | 2 | 13 | 1.34 | 3.7 | 0.060 | 0.530 | | | | | |
| | 3 | 4 | 2.43 | 14.0 | 0.031 | 0.429 | | | | | |
| | 4 | 1 | 4.97 | 53.2 | 0.012 | 0.347 | | | | | |
| GIS Results | | | | | | | 3.30 | 2.60 | 4 | 0.60 | 1.10 |
| %Error | | | | | | | 9.4 | 13.7 | 5.0 | 13.3 | 9.1 |
| 3. Kasilian | 1 | 49 | 0.49 | 1.1 | 0.109 | 0.563 | 3.65 | 2.09 | 3.92 | 0.72 | 1.3 |
| | 2 | 13 | 1.03 | 4.4 | 0.073 | 0.436 | | | | | |
| | 3 | 4 | 2.19 | 17.3 | 0.038 | 0.337 | | | | | |
| | 4 | 1 | 4.65 | 67.8 | 0.008 | 0.261 | | | | | |
| GIS Results | | | | | | | 3.5 | 1.5 | 4.3 | 0.4 | 1.1 |
| %Error | | | | | | | 4.3 | 43.2 | 8.8 | 89.5 | 18.2 |

Figs 6, 8 depict the GIS based and computational GP concerning the three case study
catchments.

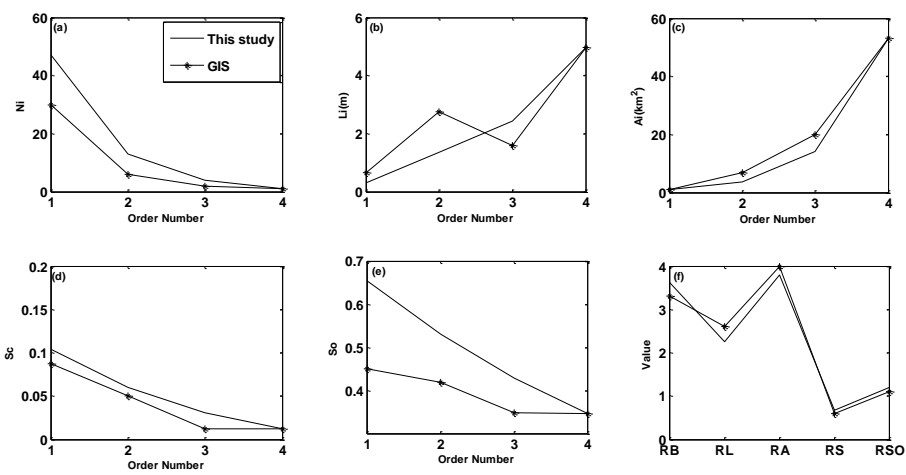




339                  Fig.7. Verification of GP in Heng-Chi catchment

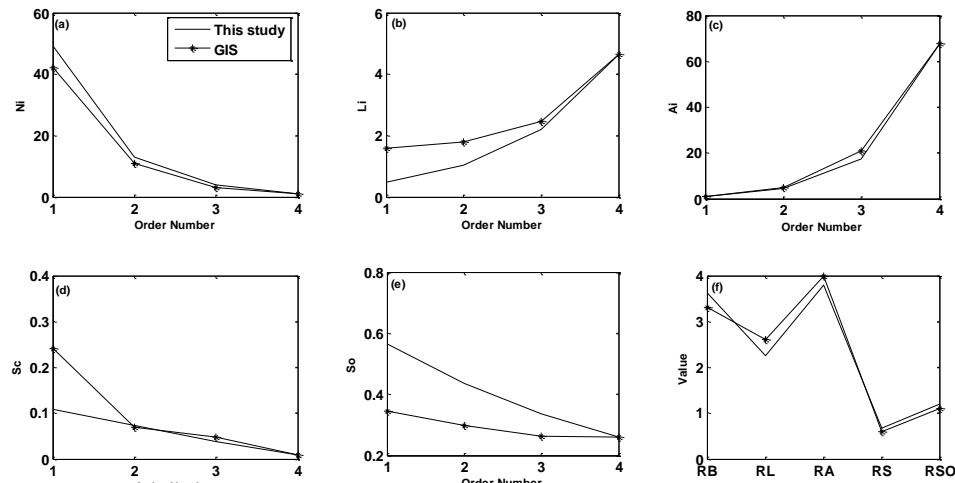


341                  Fig.8. Verification of GP in Kasilain catchment

The mean errors of regression equations in estimation of $R_B$, $R_L$, $R_A$, $R_S$, and $R_{SO}$ in the three
selected catchments are, respectively, 4.7%, 23.5%, 7.1%, 41.3%, and 22.9%.
The greatest errors of the model emerged in estimation of, respectively, $R_S$, $R_L$, $R_{SO}$, $R_A$, and $R_B$.
As observed in Fig. (5a), the stream slope ratio has a slight affect on runoff, so its error could be
ignored. Regarding high sensitivity of the length and overland slope ratios their errors range
from 23 to 24 percent and it is recommended that the joint effects of all the ratios on DRH of the
selected catchments be considered.
In the previous sections, the influences of GP on runoff were pondered separately, and the GP of
the three catchments were estimated via the regression equations. To study accuracy of the
estimations more deeply it is better to estimate the DRH using GIUH model. For this purpose,
taking the information about excess rainfall hyetograph and recorded runoff of the Kasilian and
the Heng-Chi catchments into consideration, we turn to verification of the predicted runoff for
the two catchments.
The model GIUH was employed in two cases, one in which geomorphologic parameters are GIS
based and the other where empirical regression equations (GIS-unsupported) are concerned for




the Kasilian and the Heng-Chi catchments. The results of the model in each case were compared
with those of observed runoff recorded. Since the observed runoff and rainfall data of Gagas
catchment were not available, this catchment was dispensed in verification phase. Figure 9
shows the results of GIUH model for DRH estimation in Kasilian catchment for two events on
10[th] May 1992 and 4[th] May1993. Also, Fig. (10) illustrates those in Heng-Chi catchment for two
events July 1996 and October 2000.

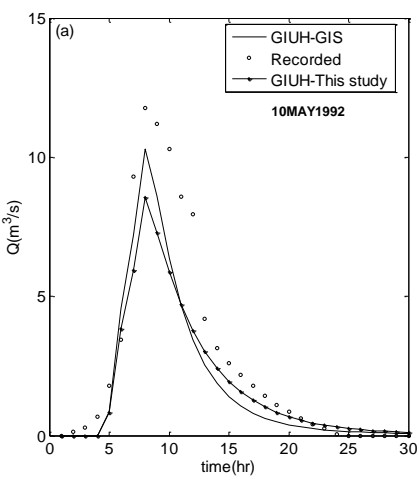
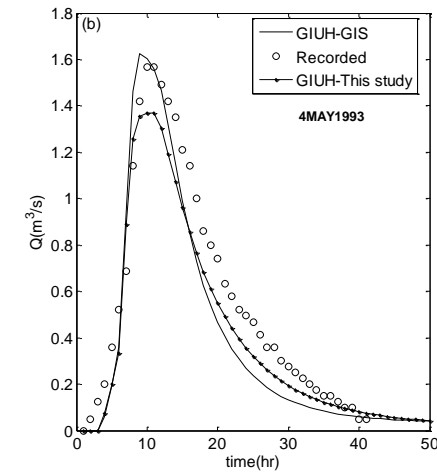


Fig.9. Estimation of Kasilian DRH by GIUH model                364

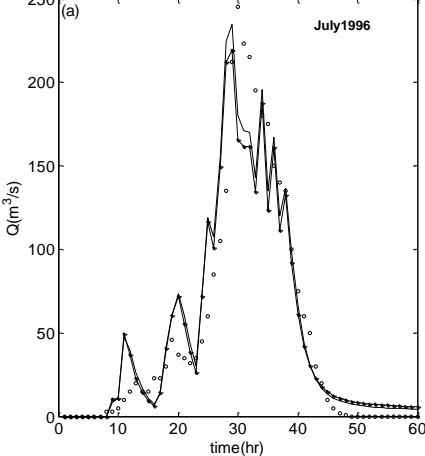
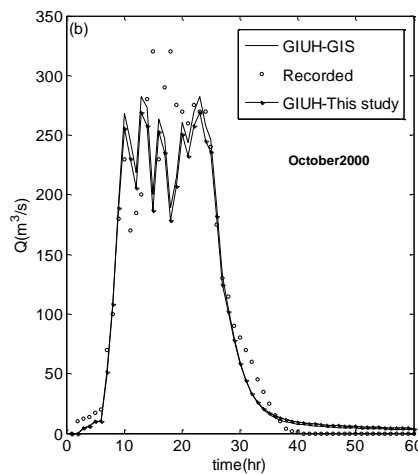






Fig.10. Estimation of Heng-Chi DRH by GIUH model          366

To validate the fitness of the model for the Kasilian and Heng-Chi catchments, three common
statistical measures were used, the coefficient of efficiency (*CE*), Root mean square error
(*RMSE*), and Relative error in peak (*REP*).
Estimation of these three parameters is carried out by the following equations:
$$CE = 1 - \frac{\sum_{t=1}^{n}[Q_r - Q_s]^2}{\sum_{t=1}^{n}[Q_r - \overline{Q_r}]^2}$$   (20)
$$RMSE = \left[\frac{1}{n}\sum_{i=1}^{n}(Q_r - Q_s)^2\right]^{0.5}$$   (21)

$$REP = 100 \times [Q_{p_s} - Q_{p_r}] / Q_{p_r}$$   (22)
where $Q_r$ is the recorded discharge at time $t$; $Q_s$ is the simulated discharge at time $t$; $\overline{Q_r}$ is the
mean recorded discharge during the storm event; $n$ is the number of discharge records during the
storm event; $Q_{p_s}$ is the peak discharge of the simulated hydrograph and $Q_{p_r}$ is the recorded
peak discharge.
Table (3) gives the values of *REP*, *CE*, and *RMSE* calculated for the two selected catchments in
GIS-supported and GIS-unsupported (this study) cases.

Table 3. Validation result of the GIUH model          381

| July 1996 | *REP*% | *CE* | *RMSE* |
|---|---|---|---|
| GIS | 4.18 | 0.87 | 24.54 |
| This study | 10.62 | 0.86 | 25.44 |
| October 2000 | | | |
| GIS | 11.81 | 0.93 | 31.22 |
| This study | 15.99 | 0.92 | 32.25 |
| 10 May 1992 | | | |
| GIS | 12.68 | 0.81 | 1.13 |
| This study | 27.33 | 0.76 | 1.26 |
| 4 May 1993 | | | |





| | | | |
|---|---|---|---|
| GIS | 3.5 | 0.87 | 0.10 |
| This study | 12.6 | 0.91 | 0.10 |


It is concluded that the computational error values of runoff peak (*REP*%) that could be inferred
(in this study) for the four rainfall-runoff events are, on average, 10% more than the error
resulting from actual information (GIS support). As seen in Figs (9) and (10), the results of the
GIUH model in the two cases concerning GIS and empirical equations are very close to each
other. *CE* and *RMSE* are near-valued as well. The mean *CE* of the model was computed for the
four events as 0.87 which is a satisfactory value.
**8. Summary and conclusion**
In this research, experimental equations were presented to work out geomorphologic parameters
of watersheds of less than 600 km$^2$ area. These equations are offered in accordance with the
nonlinear regression method fitted to the geomorphologic parameters of nine different
catchments of the world. The equations were taken under verification in three other selected
catchments, and their results were compared with those calculated from GIS. Finally, direct
runoff hydrograph was estimated by GIUH with regard to the geomorphologic data computed for
the three catchments, and then compared to the observed values. Sensitivity of bifurcation ratio,
length ratio, area ratio, stream slope ratio, and overland slope ratio to runoff of Kasilian
catchment were investigated. It is shown that the relative sensitivity of $R_B$, $R_L$, $R_A$, $R_S$, and $R_{SO}$
was 0.56, 0.01, 0.92, 0.042, and 1.33, respectively. The greatest effect was related to,
respectively, the overland slope ratio, length ratio, and bifurcation ratio, and the least effect was
related to area ratio, and streams slope ratio.
The geomorphologic parameters of three catchments Gagas, Heng-Chi, and Kasilian were
determined based on the experimental equations given in this research, and compared with their
actual results. The average errors of the model in estimation of $R_B$, $R_L$, $R_A$, $R_S$, and $R_{SO}$ in the
three case study catchments were 4.7%, 23.5%, 7.1%. 41.3%, and 22.9%, respectively.
Lastly, the estimated geomorphologic parameters was input into the GIUH model and the values
of direct runoff hydrograph of two catchments Kasilian and Heng-Chi were calculated and
compared with those of observed runoff. According to the results, the computational error values





of runoff peak (*REP*%) for the four rainfall-runoff events are, on average, 10% more than the
error resulting from actual information (GIS-Supported). The results of the GIUH model in the
two cases concerning GIS and without GIS are very close to each other. *CE* and *RMSE* in the
two cases are near-valued as well. The mean coefficient of efficiency of the model was computed
for the four events as equal to 0.87.

**9-Acknowledgements:** The authors would like to thank Estahban Branch, Islamic Azad
University for the financial support of this research, which is based on a research project
contract.

## 10-References 418

Alemngus, A, Mathur, B.S (2014) GEOMORPHOLOGIC INSTANTANEOUS UNIT
HYDROGRAPHS FOR RIVERS IN ERITREA (EAST AFRICA), Journal of Indian Water
Resources Society, Vol. 34, No. 1.
Choi, Y, Lee, G, Kim, J (2011). Estimation of the Nash model parameter based on concept of
geomorphologicdispersion. J HydrolEng 16(10), 806–817.
Chang, C.-H and Lee, K. T (2008) Analysis of geomorphologic and hydrological characteristics
in watershed saturated areas using topographic-index threshold and geomorphology-based runoff
model, Hydrol. Process., 22, 802–812.
Gupta, V.K, Waymire, E and Wang, C.T (1980) A representation of an instantaneous unit
hydrograph from geomorphology. Water Resour. Res. 16 (5), 855–862.
Horton, R. E (1932) Drainage-basin characteristics, Eos Trans. AGU, 13,350– 361.
Horton, R. E (1945) Erosional development of streams and their drainage basins: Hydrophysical
approach to quantitative morphology, Geol. Soc.Am. Bull., 56, 275– 370.
Kumar, R, Chatterjee, C , Singh, RD, Lohani, AK and Kumar, S (2004) GIUH based Clark and
Nash models for runoffestimation for an ungauged basin and their uncertainty analysis. Int J
River Basin Manag, 2(4):281−190.
Kumar, R, Chatterjee, C, Singh, RD, Lohani, AK and Kumar, S (2007) Runoff estimation for an
ungauged catchmentusing geomorphologic instantaneous unit hydrograph (GIUH) models.
Hydrol Process 21(14):1829−1840.





Kumar, A, Kumar, D (2008) Predicting direct runoff from hilly watershed using geomorphology
and stream-order law ratios: case study. J Hydrol Eng, 13(7), 570–576.
Kumar, A (2015) Geomorphologic Instantaneous Unit Hydrograph Based Hydrologic Response
Models for Ungauged Hilly Watersheds in India, Water Resources Management, February, Vol
442 29, 3, 863-883.

Lee, K.T., Yen, B.C., 1997. Geomorphology and kinematic-wave based hydrograph derivation.
J. Hydrol. Eng. ASCE, 123 (1), 73–80.
Lee, K.T (1998) Generating design hydrographs by DEM assisted geomorphic runoff simulation:
a case study. J. Am. Water Resour. Assoc. 34 (2), 375–384.
Lee, K. T, and Chang, C. H (2005) Incorporating subsurface-flow mechanism into
geomorphology-based IUH modeling. Journal of Hydrology, 311:91–105.
Rodriguez-Iturbe, I, Valdes, J.B (1979) The geomorphologic structure of hydrologic response.
Water Resour. Res. 15 (6),1409–1420.
Rodriguez-Iturbe, I, Gonzalez-Sanabria, M, Bras, R.L (1982). Ageomorphoclimatic theory of the
instantaneous unit hydrograph. Water Resour. Res. 18 (4), 877–886.
Smart, J. S (1972) Channel networks., Advances in hydroscience, Vol.8, V. T. Chow, ed.,
Academic Press, Inc., San Diego, Calif., 305-346.
Strahler, A. N (1952) Hypsometric (area-altitude) analysis of erosionaltopology, Bull. Geol. Soc.
Am., 63, 1117–1142.
Strahler, A. N (1957)  Quantitative analysis of watershed geomorphology, Trans. AGU, 38(6),
913– 920.
Strahler, A. N (1964) Quantitative geomorphology of drainage basins and channel networks, in
Handbook of Applied Hydrology, edited by Ven te Chow, pp. 4 –39, McGraw-Hill, New York.
Sabzevari, T, Fattahi, M.H, Mohammadpour, R and  Noroozpour, S (2013) Prediction of surface
and subsurface flow in catchments using the GIUH, under publication. Journal of Flood Risk
Management. , Vol 6. Issue 2, 135–145.
Sabzevari, T, Noroozpour, S (2014) Effects of hillslope geometry on surface and subsurface
flows, Hydrogeology Journal,, Vol22,7, 1593-1604.



Shadeed, S, Shaheen, H and Jayyousi, A (2007) GIS-BASED KW–GIUH  HYDROLOGICAL
MODEL OF SEMIARID CATCHMENTS: THE CASE OF FARIA CATCHMENT,
PALESTINE, The Arabian Journal for Science and Engineering, Volume 32, Number 1C.
Shuyou, C,  Lee, K.T,  Juiyi H,  Xingnian, L,  Huang, E and  Yang, K (2010) Analysis of Runoff
in Ungauged Mountain Watersheds in Sichuan, China using Kinematic-wave-based GIUH
Model, Journal of Mountain Science, Vol 7,2 , 157-166.
Yen, B. C, and Lee, K. T (1997) Unit hydrograph derivation for ungauged watersheds by stream-
order laws, J. Hydrol. Eng., 2(1), 1–9.