# Peer review of "Prediction of geomorphologic parameters of catchment without GIS to"

_Hydrology and Earth System Sciences, 2016_

## Referee Comment (RC1) · Anonymous Referee #1 · 14 May 2016

General comments: Prediction in ungauged catchments has great importance, and the geomorphologic instantaneous unit hydrograph (GIUH) is a powerful tool in such ungauged catchments on the procedure of runoff concentration. In this paper, the authors presented an idea to compute geomorphologic parameter (GP) which be used in GIUH without the need of GIS or DEM. Some regression equations were presented on the basis of nine natural catchments in different countries and evaluated in the other three different catchments. And then, the sensitivity of different GP on direct runoff hydrograph was analyzed. Finally, the effects of GIUH by means of GP from regression equations were examined. The method in this paper is interesting.

Specific comments: In general, high resolution DEM could be obtained conveniently

and the GP could be calculated easily by means of some GIS software (for example the 'River Tools'). The purpose of this paper is to calculate GP without DEM, how could we obtained the catchment area (A) and the length of main stream (L)? From the Fig.9 and Fig. 10 we could find that in the four events, calculated peak flow was lower than observed peak flow for three events. Is these mean that some other factors affected the GIUH? Page 6 Line 133-134ïijŽWhat do the characters "a" and "b" denote? Page 11 Line 219-221: When the Eq.(12) indicates it can be applied in small catchment which is less than 200km2; how can we know it can be used in those watersheds beneath 600 km2?

Technical corrections: Page 19 Fig.10 (a): The legend was omitted.

---

## Author Comment (AC1) · 21 May 2016

Answers to referee 1

Dear reviewer, Thank you very much for your attention and response. we have corrected our paper ("Prediction of geomorphologic parameters of catchment without GIS to estimate runoff using GIUH model"). In the following, you can find our responses.

Question1: The purpose of this paper is to calculate GP without DEM, how could we obtained the catchment area (A) and the length of main stream (L)? Answer 1: In order to create a suitable digital elevation model (DEM) high quality aerial maps or topographic maps are needed. Since the preparation of such maps is usually costly,

there are many watersheds that lack DEMs. However, for most urban and rural areas in the world 1:25000-scale topographic maps now exist so that the watershed boundary can be delineated and consequently the values of watershed area and main stream length can be estimated with a good approximation.

Question2: From the Fig.9 and Fig. 10 we could find that in the four events, calculated peak flow was lower than Observed peak flow for three events. Is these mean that some other factors affected the GIUH?

Answer 2: The key question of the respected reviewer can be applied to any other rainfall-runoff model. It is clear that there has been no model considering all effective factors on rainfall-runoff transformation. Regarding our particular model, most GIUH models need excess rainfall as an input. Infiltration parameters values are different from one event to another. The computation of excess rainfall is quit complex, therefore the resultant errors can also affect the model results. In our study all watershed parameters were assumed to be constant and given this assumption geomorphologic parameters were calculated. Additionally, we tested our model for four events. Our results, in our opinion, do not mean that the model will underestimate the peak discharge for any other cases.

Question3: Page 6 Line 133-134 What do the characters "a" and "b" denote? Answer 3: Parameters a and b are constants of Eq. 4.

Question4: Page 11 Line 219-221: When the Eq.(12) indicates it can be applied in small catchment which is less than 200km2 ; how can we know it can be used in those watersheds beneath 600 km2 ? Answer4: To derive the regression equation (Eq. (12)) we used the information collected from 80 watersheds with an area ranging from 1 to 600 km2. Accordingly, it is recommended that the equation be used for the watersheds smaller than 600 km2. In addition, line (219-221) was revised as follows:

"Eq. (12) indicates that in small catchments with area less than 600km2, the value of RB runs between 3.47 and 4. It is suggested that Eq. (12) be applied to catchments of areas beneath 600km2."

Question5: Technical corrections: Page 19 Fig.10 (a): The legend was omitted
Answer5: Fig. 10(a) was completed as below:

Please also note the supplement to this comment:
http://www.hydrol-earth-syst-sci-discuss.net/hess-2016-153/hess-2016-153-AC1-
supplement.pdf

**Supplement:**

[revised manuscript text omitted]

---

## Referee Comment (RC2) · Anonymous Referee #2 · 17 Jun 2016

This paper introduced an interesting (but not novel) idea to estimate the stream-order-law ratios and Geomorphologic parameters (GP) in ungauged basins for the application of GIUH model. Regression equations of stream-order-law ratios and GP were firstly developed in nine catchments and then evaluated in other three catchments. The performances of GIUH model using estimated and calculated stream-order-law ratios were compared in term of simulating runoff events. However, I find this manuscript in some aspects rather poor. From a more scientifically and technically rigorous perspective, this manuscript needs to be significantly improved. One of my major concerns is that the manuscript is not well written, with lots of ill-formulated sentences that are not easy to follow. Please keep in mind that quality is paramount, schedule is secondary, and

reviewers and editors really hope authors improve the quality of manuscripts as best as they can before submission. I think the authors even did not read the manuscript thoroughly before submission. It looks like the paper was finished in a short time and submitted hastily. Another major concern is about the scientific merits of this work. First, I do not agree that digital elevation model (DEM) is not available in most catchments. For example, the Shuttle Radar Topography Mission (SRTM) provides global elevation maps in a resolution of 3 arc second (even 1 arc second in some countries), which has been widely used for hydrological application. The products can be downloaded without charge from websites such as http://srtm.csi.cgiar.org/. Second, the authors repeatedly reiterated that it takes a long time to compute stream-order-law ratios based on DEM with GIS. My question is how long time do you think it is too long to wait, one day, one week or one month? The study catchments are all smaller than 600 km2 in this work. Computing stream-order-law ratios with GIS for catchments with such a small area could take only few hours. Considering stream-order-law ratios are constant at long time scale, we do not need to re-compute their values when using GIUH in various time periods. The results also indicate a 10% larger error for peak flow obtained by the proposed method, then why not just wait for few hours (or days) to reduce this error. The authors also highlighted that some stream-order-law ratios should be calculated manually by GIS users. Actually, most GIS software (e.g., ArcGis) are open platforms to develop new toolbox. GIS users can write scripts (in Python or in Arc workstation) to calculate the related variables automatically, and transfer to following users. Laziness is not a sufficient reason to reduce the accuracy of model simulation.

Special comments: 1. Line 38-40: Why are these sentences here? How do these sentences relate to the former and later paragraphs? 2. Line 45: You have 'first' here, then where is 'second'? 3. Line 49: You introduce the calculation of GP using GIS in former sentences, but here you go back to the application of GIUH. 4. Line 54-76: Application of GIUH model is not the main point of this work. The main point focuses on the estimation of stream-order-law ratios when using GIUH. Hence, there is no necessary to list so many references on the application of GIUH. But please

provide more references about the estimation of stream-order-law ratios in previous application of GIUH. If I am correct, the GIUH model was proposed by Rodriguez-Iturbe and Valdes in 1979, while the first study to calculate stream-order-law ratios based on DEM was introduced by Lee (1998). What methods were used to estimate stream-order-law ratios during 1979-1998? Are there any methods similar to the proposed one in this work? 5. Line 77-85: This paragraph needs re-organization, taking my second major concern into account. Moreover, you list many disadvantages of the GIS-based method, but not any advantages. If it is so time-consuming, then why someone used the GIS-based method in their areas? Also, please provide a review on the application of GIS-based method in existing literatures. 6. Line 86: What is the full name of SPSS software? Can you give some geostatistical references that use this software as well? The question is why use SPSS? 7. Line 95: It is abrupt to see this sentence here, without any related expression before. Why should analyze the sensitivity of stream ratios? What the relation between this topic and the main story of this work? 8. Section 3: It looks like you are introducing the equations for stream-order-law ratios, not geomorphologic parameters. 9. Section 3: Listing these equations one by one makes the manuscript too prolix to read. You can summarize them in one table. 10. Line 176-179: There is no necessary to repeat the ideas again. Moreover, these words are hard to follow. 11. Section 4: You just introduce the study catchments in this section, it is not 'case study'. It is better to summarize all study catchments in Table 1, including the size, reference, and geographical extent. 12. Figure 1: Why only Kasilian and Gagas? If you do not have the map for Heng-Chi, please delineate it from the SRTM DEM. The maps of Kasilian and Gagas could be originally delineated from DEM as well. 13. Line 212-216: What are the 80 watersheds and 37 catchments mentioned here? You told that you obtained these equations based on nine catchments listed in Table 1. 14. Line 212-216: What is the input data for the software? Did you use the calculated stream-order-law ratios from DEM to determine the coefficients in the equations? If yes, then DEM data is needed to apply your method in other catchments, as you cannot expect that these equations can be applicable to any catchments in the

world. 15. Section 5: Same to comment 9, there is no need to list the equations one by one, but summarize in one table. 16. Section 5: So many assumptions adopted here. Are there any references to support your assumptions? 17. Section 5: You just write out these equations followed by correlation coefficients, without any figures to show the match between the estimated and GIS-based stream-order-law ratios and GP. I really doubt the credibility of the correlation coefficients. 18. Line 252: What is "simple topographic maps of the catchment", do you think this kind of map are of course available for most catchments? My concern here is how to calculate the area and river length for the application of your regression equations in other places. 19. Line 252-262: Add a sub-section 5.6 for these words. 20. Line 252: You often have a mixed use between subcatchment and catchment. Please check the manuscript thoroughly and make sure all the using are appropriate. 21. Line 328: The word 'observed' is not appropriate here. How to observe the stream ratios? 22. Section 7: There are two points in this section. One is the comparison between estimated and GIS-based GP. The other one is the comparison of GIUH performance using estimated and GIS-based stream-order-law ratios. I think it is better to divide this section into two sub-sections. 23. Table 2: How to calculate the Error here? 24. Line 409: How did you calculate the average error? 25. Captions for tables and figures are too simple to explain the implied information. Give full names of all the abbreviations in captions. 26. I do not like the structure of this manuscript. It is a bit fragmented. The authors should develop a 'Methodology' section to summarize the work procedure, and introduce the estimation of regression equations, the verification procedure and the calculation of metrics in more details, without repeating in following sections. In general, this paper is far away from publishable in HESS from my taste. The structure is fragmented, the English writing is wordy with lots of the ill-formulated sentences. My major concern is that the scientific contribution of the proposed idea to the application of GIUH in ungauged basins should be rather limited, considering the widely and easily using of DEM data and GIS software.

---

## Author Comment (AC2) · 18 Jul 2016

Answers to referee 1

Dear Reviewer, Thanks for your efficient review. The comments were effective and lead to more scientific and well written paper. The structure of the paper has been reviewed and some reforms have been applied. The writing format has been reviewed and some changes were made. In particular, some major corrections were done in the introduction of the manuscript. The writing of the manuscript was checked and corrected by an English native speaker. The requested corrections applied and the revised version of the manuscript was sent to the referee. We tried to make our view to GIS and preparing DEM more accurate. To this end after consulting with GIS experts

some changes in key words according to referee viewpoint applied. The reforms are as following:

Question1: One of my major concerns is that the manuscript is not well written, with lots of ill-formulated sentences that are not easy to follow. Please keep in mind that quality is paramount, schedule is secondary, and reviewers and editors really hope authors improve the quality of manuscripts as best as they can before submission. Answer 1: The writing of the paper has been reviewed and corrected by English language experts. Ill-formatted sentences were recognized and revised. Undoubtedly our corrections could be done more efficiently if these sorts of paragraphs have been identified by reviewer.

Question2: Another major concern is about the scientific merits of this work. First, I do not agree that digital elevation model (DEM) is not available in most catchments. For example, the Shuttle Radar Topography Mission (SRTM) provides global elevation maps in a resolution of 3 arc second (even 1 arc second in some countries), which has been widely used for hydrological application. The products can be downloaded without charge from websites such as http://srtm.csi.cgiar.org/.

Answer 2:

We agree with you in this part that some websites such as SRTM provides DEM for all parts of the world but there is a question about their quality for hydrologists. There are different ideas in this regard and our purpose in this research based on this reality that: "there are some catchments with non-qualified DEM" although we accept that we could not explain this purpose clearly and then the related sentences were corrected. For confirmation of this issue we provide some resources that do not accept the 90 meter's DEMs for estimation of runoff.

"Hancock, G. R., Martinez, C., Evans, K. G. and Moliere, D. R. (2006), A comparison of SRTM and high-resolution digital elevation models and their use in catchment geomorphology and hydrology: Australian examples. Earth Surf. Process. Landforms, 31:

1394–1412. doi: 10.1002/esp.1335"

"Rahman, M. M., Arya, D. S., Goel, N. K., Limitation of 90 m SRTM DEM in drainage network delineation using D8 method—a case study in flat terrain of Bangladesh, Applied Geomatics , (2010) 2:49–58."

It goes without saying that geomorphologic information obtained from high resolution DEM leads to more accurate runoff estimation. In this research the results obtained from GIS were considered as reference and the results of proposed method were compared with GIS results. Some of hydrologists are not familiar with GIS and related software and don't want to pay for using GIS experts and they are tend to apply easier rainfall-runoff models. The aim of this research is to provide another method to estimate geomorphologic information easier even not so accurate. To sum up, the proposed method is a new method which should be applied, evaluated and verified in more catchments in future.

Question3: Second, the authors repeatedly reiterated that it takes a long time to compute stream-order-law ratios based on DEM with GIS. My question is how long time do you think it is too long to wait, one day, one week or one month? The study catchments are all smaller than 600 km2 in this work. Computing stream-order-law ratios with GIS for catchments with such a small area could take only few hours. Answer 3: The referee idea in this regard has been applied and the repeated sentence has been changed but we should mention that GIS experts in my country prefer the DEMs provided from 1/25000 topography maps to the DEMs obtained from some websites. This issue is based on their experiences and we don't want to accept or reject it but we should accept that providing high quality DEMs and then preparing geomorphologic information and sometimes writing scripts in GIS are time consuming and as told before it is not preferred by many hydrologists. We should mention again that cost and time for providing information are not our main point in this research but we want to present another method for providing geomorphologic information and stream order law for using in GIUH models.

Question4: The authors also highlighted that some stream-order-law ratios should be calculated manually by GIS users. Actually, most GIS software (e.g., ArcGis) are open platforms to develop new toolbox. GIS users can write scripts (in Python or in Arc workstation) to calculate the related variables automatically, and transfer to following users. Laziness is not a sufficient reason to reduce the accuracy of model simulation.

Answer 4: I completely agree with the referee opinion but as it was told before, many of hydrologists and GIS users are not familiar with writing scripts (in Python or in Arc workstation). Therefore, through this research we wanted to present a more user-friendly method.

Special comments:

Question5: Line 38-40: Why are these sentences here? How do these sentences relate to the former and later paragraphs? Answer 5: The structure of introduction has been changed and the position of intended sentences has been revised.

Question6: Line 45: You have 'first' here, then where is 'second'? Answer 6: It has been corrected.

Question7: Line 49: You introduce the calculation of GP using GIS in former sentences, but here you go back to the application of GIUH Answer 7: It has been corrected in the manuscript.

Question8: Line54-76: Application of GIUH model is not the main point of this work. The main point focuses on the estimation of stream-order-law ratios when using GIUH. Hence, there is no necessary to list so many references on the application of GIUH. But please provide more references about the estimation of stream-order-law ratios in previous application of GIUH. If I am correct, the GIUH model was proposed by Rodriguez-Iturbe and Valdes in 1979, while the first study to calculate stream-order-law ratios based on DEM was introduced by Lee (1998). What methods were used to estimate streamorder- law ratios during 1979-1998? Are there any methods similar

to the proposed one in this work? Answer 8: Many searches about other methods for estimation of stream-order-haw ratios and gemorphologic data have been done but almost all of the investigations were about estimation of these ratios using GIS. During years 1979 – 1998 some researches were found which were added. Some references about the estimation of these ratios were provided as following:

"Studies on streams orderings of catchments were first introduced by Horton (1932, 1945). Later, modifications were made on Horton's method by Strahler (1952, 1957, 1964) leading to a new method of ordering. Sherve (1966) concluded that the Stahler stream numbers generally gave a better fit for natural stram networks than did the Horton stream numbers. Horton-Strahler's laws were extensively used in geomorphological applications to classify river systems [e.g., Raff et al., 2003; Reis, 2006], to establish relations with the fractal nature of channel network as detailed by Rodríguez-Iturbe and Rinaldo [1997] [e.g., Beer and Borgas, 1993; La Barbera and Roth, 1994; Rodríguez-Iturbe et al., 1994], and to characterize scale properties [Claps et al., 1996; Peckham and Gupta, 1999; Veitzer and Gupta, 2000; Dodds and Rothman, 1999, 2001]." "Using GIS tool is still one of best ways of calculating the geomorphologic parameters (Sarangi et al.2003; Obi Reddyet al.2004; Valeriano et al. 2006; Ozdemir and Bird, 2009)"

Question9: Line 77-85: This paragraph needs re-organization, taking my second major concern into account. Moreover, you list many disadvantages of the GIS-based method, but not any advantages. If it is so time-consuming, then why someone used the GIS-based method in their areas? Also, please provide a review on the application of GIS-based method in existing literatures. Answer 9: The intended sentences were corrected. Some references about using GIS in geomorphologic ratio calculation were added.

Question10: Line 86: What is the full name of SPSS software? Can you give some geostatistical references that use this software as well? The question is why use SPSS? Answer 10: The full name of SPSS (statistical package for the social sciences) has been added. This statistical software is famous but we added two references for application of SPSS as below: "Mohamoud, Y. M., and Parmar, R. S. (2006): ESTIMATING STREAMFLOW AND ASSOCIATED HYDRAULIC GEOMETRY, THE MID-ATLANTIC REGION, USA1. Journal of the American Water Resources Association, 42(3), 755." "Norusis, M.J. (1999): SPSS regression models 10.0." Since the SPSS is a quit simple and commonly-used we used it to perform statistical analysis.

Question11: Line 95: It is abrupt to see this sentence here, without any related expression before. Why should analyze the sensitivity of stream ratios? What the relation between this topic and the main story of this work? Answer 11: We agree. That sentence has been deleted.

Question12: Section 3: It looks like you are introducing the equations for stream-order-law ratios, not geomorphologic parameters. Answer 12: We agree. The title of section 3 has been changed to "Horton-Strahler stream-order-law ratios "

Question13: Section 3: Listing these equations one by one makes the manuscript too prolix to read. You can summarize them in one table Answer 13: The equations of this chapter were summarized in a table 1 (In revised manuscript).

Question14: Line 176-179: There is no necessary to repeat the ideas again. Moreover, these words are hard to follow Answer 14: The sentence has been deleted.

Question15: Section 4: You just introduce the study catchments in this section, it is not 'case study'. It is better to summarize all study catchments in Table 1,including the size, reference, and geographical extent. Answer 15: I agree. The title of "CASE STUDY" has been deleted and related information transferred to table 2.

Question16: Figure 1: Why only Kasilian and Gagas? If you do not have the map for Heng-Chi, please delineate it from the SRTM DEM. The maps of Kasilian and Gagas could be originally delineated from DEM as well. Answer 16: The map of Heng-chi has been added as figure1(In revised manuscript).

Question17: Line 212-216: What are the 80 watersheds and 37 catchments mentioned

here? You told that you obtained these equations based on nine catchments listed in Table 1. Answer 17: Relation between catchment area and bifurcation ratio (Eq. 12) was obtained from information of 37 catchments. Equations 13 to 16 were derived based the information of 9 catchments because there was not all needed information in the 37 catchments.

Question18: Line 212-216: What is the input data for the software? Did you use the calculated stream-order-law ratios from DEM to determine the coefficients in the equations? If yes, then DEM data is needed to apply your method in other catchments, as you cannot expect that these equations can be applicable to any catchments in the world

Answer 18: In equation 12 bifurcation ratio obtained based on catchment area. For this purpose area (as input data) and Rb information of 37 catchment (as output data) have been used and regression equation as follow has been obtained. Figure2 added in this part to show linear regression between bifurcation ratio and the area of the catchment.

If catchment area is available, Rb obtained from the regression equation and DEM is not needed. Eqs. 12 to 16 the SOLR coefficients of which were estimated using their DEMs are first derived based on the data of the representative catchments.Then the equations are applied to the catchments without DEM. Given Eq. 12 and the catchment area coefficient RB is first estimated. Next coefficient RL is calculated from Eq. 13 using the length of the main stream and catchment area. Then the value of RA is obtained from Eq. 14. Finally Eqs. 15 and 16 give the values of Rs and Rso.

Question19: Section 5: Same to comment 9, there is no need to list the equations one by one, but summarize in one table Answer 19: Each equation needs to be referred and described individually so the title of parts related to equations have been omitted but according to our opinion it is better not to put these equations in a table.

Question20: Section 5: So many assumptions adopted here. Are there any references to support your assumptions? Answer 20: Some references were introduced for assumptions in part 4.

Question21: Section 5: You just write out these equations followed by correlation coefficients, without any figures to show the match between the estimated and GIS-based stream-order-law ratios and GP. I really doubt the credibility of the correlation coefficients. Answer 21: You are correct. Figure3 (In revised manuscript) shows the GIS-based stream-order-law ratios versus predicted SOLR

Question22: Line 252: What is "simple topographic maps of the catchment", do you think this kind of map are of course available for most catchments? My concern here is how to calculate the area and river length for the application of your regression equations in other places

Answer 22: By simple topographic maps of the catchment we meant topographic catchments maps (Scale 1:25000 to 1:50000). The word "simple" which seemed confusing was deleted from the text. Usually topographic maps with mentioned scales are available for all catchments and the first step of each hydrologist work is catchment delineation according to these maps and then calculating catchment area and length.

Question23: Line 252-262: Add a sub-section 5.6 for these words. Answer 23: A section with this title "5. Prediction of catchment's geomorpological information" added.

Question24: Line 252: You often have a mixed use between subcatchment and catchment. Please check the manuscript thoroughly and make sure all the using are appropriate. Answer 24: This issue has been reviewed and corrected. Our purpose in figure 4 were sub-catchment.

Question25: Line 328: The word 'observed' is not appropriate here. How to observe the stream ratios? Answer 25: I agree. The word "observed" has been deleted.

Question26: Section 7: There are two points in this section. One is the comparison between estimated and GIS-based GP. The other one is the comparison of GIUH performance using estimated and GIS-based stream-order-law ratios. I think it is better

to divide this section into two sub-sections. 23. Table 2: How to calculate the Error here? Answer 26: Validation part has been divided to two sections: 7.1. Validation of stream-order-law ratios 7.2. Validation of catchment's direct runoff

Question27: Line 409: How did you calculate the average error? Answer 27: Simple mean has been used in 4 events and there is no need to provide the formula.

Question28: Captions for tables and figures are too simple to explain the implied information. Give full names of all the abbreviations in captions. Answer 28: This issue has been corrected.

Please also note the supplement to this comment:
http://www.hydrol-earth-syst-sci-discuss.net/hess-2016-153/hess-2016-153-AC2-supplement.zip

Fig. 1: Drainage network map
a)Heng-Chi catchment b) Gagas catchment  c) Kasilian catchment

**Fig. 1.** Fig. 1: Drainage network map a)Heng-Chi catchment b) Gagas catchment c) Kasilian catchment

RB = 0.0027A + 3.47
R² = 0.8

Fig.2: Linear regression between bifurcation ratio and the area of the catchment
**Fig. 2.** Fig.2: Linear regression between bifurcation ratio and the area of the catchment

[Figure]

Fig3: GIS-based stream-order-law ratios versus predicted SOLR

**Fig. 3.** Fig3: GIS-based stream-order-law ratios versus predicted SOLR